# Application of Machine Learning in the Quantitative Analysis of the Surface Characteristics of Highly Abundant Cytoplasmic Proteins: Toward AI-Based Biomimetics

**DOI:** 10.3390/biomimetics9030162

**Published:** 2024-03-06

**Authors:** Jooa Moon, Guanghao Hu, Tomohiro Hayashi

**Affiliations:** 1Department of Materials Science and Engineering, School of Materials and Chemical Technology, Tokyo Institute of Technology, Yokohama 226-8502, Japan; jooamoon9703@gmail.com (J.M.); mercuryhgh@gmail.com (G.H.); 2The Institute for Solid State Physics, The University of Tokyo, Kashiwa 277-0882, Japan

**Keywords:** bioinformatics, machine learning, protein surfaces, surface engineering

## Abstract

Proteins in the crowded environment of human cells have often been studied regarding nonspecific interactions, misfolding, and aggregation, which may cause cellular malfunction and disease. Specifically, proteins with high abundance are more susceptible to these issues due to the law of mass action. Therefore, the surfaces of highly abundant cytoplasmic (HAC) proteins directly exposed to the environment can exhibit specific physicochemical, structural, and geometrical characteristics that reduce nonspecific interactions and adapt to the environment. However, the quantitative relationships between the overall surface descriptors still need clarification. Here, we used machine learning to identify HAC proteins using hydrophobicity, charge, roughness, secondary structures, and B-factor from the protein surfaces and quantified the contribution of each descriptor. First, several supervised learning algorithms were compared to solve binary classification problems for the surfaces of HAC and extracellular proteins. Then, logistic regression was used for the feature importance analysis of descriptors considering model performance (80.2% accuracy and 87.6% AUC) and interpretability. The HAC proteins showed positive correlations with negatively and positively charged areas but negative correlations with hydrophobicity, the B-factor, the proportion of beta structures, roughness, and the proportion of disordered regions. Finally, the details of each descriptor could be explained concerning adaptative surface strategies of HAC proteins to regulate nonspecific interactions, protein folding, flexibility, stability, and adsorption. This study presented a novel approach using various surface descriptors to identify HAC proteins and provided quantitative design rules for the surfaces well-suited to human cellular crowded environments.

## 1. Introduction

The intracellular space of living organisms is highly crowded with macromolecules, which can occupy up to nearly one-third of the entire cellular volume [1]. The resulting highly crowded environment poses challenges of nonspecific interactions, critically influencing issues such as protein folding, stability, and adsorption [2,3,4]. In human cells, these issues are especially crucial since the intracellular proteins that fail to fold correctly into their native shapes tend to aggregate and cause cellular malfunction and death, resulting in detrimental pathological consequences [5]. In particular, cytoplasmic proteins with high abundance, i.e., highly expressed proteins, are more likely to encounter nonspecific interactions due to the law of mass action [6]. Thus, highly abundant cytoplasmic (HAC) proteins must exhibit certain physicochemical, structural, and geometrical characteristics to adapt to the environment and mitigate the issues. Eventually, intracellular proteins, especially highly abundant ones, are expected to share particular characteristics differentiated from extracellular proteins, which often experience less crowded environments [4,7], to ensure proper cellular function in such a highly crowded environment.

Previously, computational approaches aided in the characterization of intracellular proteomes, with various techniques targeting different regions of proteins, including global regions (where proteins’ characteristics are investigated across their entire amino acid lengths) [8], surface regions [9,10], or both regions [11]. Notably, the surface regions of proteins are essential for studying protein characteristics since the regions are directly exposed to the external environment and potential partners and thus reflect various properties [4,10]. While there have been several works on using the frequency of surface residues [9,10], there is a lack of research revealing quantitative relationships among specific physicochemical, structural, and geometrical descriptors, which can have different scales for characterizing the surfaces of the HAC protein.

To address this issue, we use interpretable machine learning (ML)-based approach to characterize the surfaces of HAC proteins by quantifying the contribution of the surface descriptors. Over the past few decades, ML techniques have been increasingly applied to predict protein–protein interactions [12], protein–ligand molecular docking [13], protein subcellular localization [14], and the 3D structure of proteins [15]. Despite significant advances in these areas, identifying protein surface characteristics using only a few representative physicochemical, structural, and geometrical descriptors remains challenging. This is the first study focusing on this specific task, thus revealing quantitative relationships among surface descriptors. By understanding the surface rules of HAC proteins in human cells through interpretable ML, this study will enable the development of efficient drug delivery systems by deepening our knowledge of the interactions between therapeutic nanoparticles and proteins [16]. 

In this study, we aimed to distinguish the surfaces of HAC proteins from those of extracellular proteins using binary classification algorithms. We extracted surface physicochemical, structural, and geometrical descriptors from protein surfaces to build a database and apply ML (Figure 1). As a first step of the database construction, we collected around 330 3D protein structures each for human HAC and extracellular proteins. Then, various descriptors of the protein surfaces, such as hydrophobicity, charged area, roughness, the B-factor, and the proportions of protein structures, were calculated for the collected 3D protein structures. Then, several supervised ML algorithms including K-Nearest Neighbor (KNN), Random Forest (RF), logistic regression (LR), and Support Vector Machine (SVM) were used to solve the binary classification of extracellular and HAC proteins. Based on excellent performance and high model interpretability, we selected the LR algorithm to explain the importance of each descriptor quantitatively. Namely, this study answers the following questions: (1) Can surface characteristics of HAC proteins be identified with several physicochemical, structural, and geometrical descriptors? and (2) Which descriptor contributes to the crowded environment-adaptive surface in human cells and to what extent? The LR model used in our study enabled the identification of HAC proteins, and coefficients from the LR represented the importance of each descriptor. 

## 2. Methodology

### 2.1. Protein Sample Collection

The datasets consist of two types of human proteins: human cytoplasmic proteins with high abundance and extracellular proteins. First, we collected cytoplasmic proteins with the highest abundance level from the PaxDb database, which is a collection of experimental data on protein abundance [17]. The cytoplasmic proteins that were also tagged with extracellular keywords (e.g., secreted, extracellular matrix, and extracellular space) in Uniprot were eliminated. Then, proteins in extracellular environments determined with experimental assay were collected (GO ID: 5615) [18]. Finally, 331 human extracellular proteins and 337 HAC proteins within the sequence length range of 100 to 700 were collected for analysis (see Appendix A for the list of collected proteins).

The 3D structures of a total of 668 proteins were collected through the Alphafold ver2.0 (Alphafold2) (https://alphafold.ebi.ac.uk/, accessed on 13 December 2023) protein structure prediction model [15,19]. Alphafold2 3D models provide entire protein structures, allowing for comprehensive surface analysis, in contrast to the partial structures often found in experimental Protein Data Bank (PDB) files from X-ray crystallography. Alphafold2 is known to be the top-ranked prediction model with a median global distance test score of 92.4 across all targets and 87.0 on the challenging free modeling category in the 14th CASP assessment (https://predictioncenter.org/casp14/zscores_final.cgi, accessed on 13 December 2023). Additionally, in most cases, Alphafold2′s structural prediction accuracy has reached experimental accuracy [15].

Even though the overall predictability of Alphafold2 is exceptional, not all predicted structures are suitable for analysis. Every residue from the Alphafold2 3D protein structure is given a per-residue metric, which reflects the structural model confidence called the predicted local distance difference test (pLDDT), scaling from 0 to 100. The pLDDT evaluates how well the predicted model agrees with experimental data using the local distance difference test Cα [20]. pLDDT > 90 is considered a high-accuracy cut-off, and pLDDT > 70 is regarded as a generally correct backbone prediction [21]. When the pLDDT is lower than 50, the predicted region is expected to be intrinsically disordered [22]. However, a low pLDDT score in Alphafold2 results from high residue flexibility and dynamic structure rather than “low confidence” [23]. Also, since disordered regions of proteins are involved in molecular recognition and hydrophobic interactions, it is essential to include the regions for the analysis [24]. Considering the potential interpretability difficulty from intrinsically disordered proteins, we set our cut-off value as an average pLDDT > 50 for the whole protein structure. Finally, we ensured that over 80% of extracellular and HAC proteins had average pLDDT values of over 70 (Figure 2). 

### 2.2. Calculation of Surface Descriptors

Previous studies have introduced several definitions of protein surfaces, each with different characteristics. Among them, we adopted solvent-accessible surface (SAS) and solvent-excluded surface (SES) for calculating the other descriptors (Figure 1) [25]. The SAS was calculated by rolling probe spheres that had an equivalent size to water molecules. We used SAS for the residue-based analysis: we assumed that a specific residue in a protein could have a maximum SAS when its neighboring amino acids were Glycines (i.e., having a Gly–residue–Gly structure). When the proportion of an actual SAS for a residue to the maximum SAS was higher than or equal to 30%, the residue was defined as a surface residue. Another protein surface used in the analysis was SES, also called the Connolly surface [26]. The surface moves inward from the SAS by a distance identical to the probe sphere radius (Figure 1). Lewis et al. discovered that this continuous and functional surface is particularly useful in calculating protein surface roughness. Then, protein surface descriptors representing various physicochemical, structural, and geometrical descriptors were calculated (Table 1) based on the two surface types. All the descriptors were computed using Python 3.9.12.

The surface hydrophobicity, charge, secondary structures, and overall morphology of proteins are critical parameters for protein structures. The normalized consensus hydrophobicity scale was used to quantitatively measure the average protein surface hydrophobicity [27]. Surface charge-related descriptors were collected by calculating the fraction of the SAS of negatively charged and positively charged amino acids under physiological conditions (pH = 7). Each surface amino acid contributing to the secondary structure was directly extracted by Pymol (http://www.pymol.org, accessed on 13 December 2023) to calculate the surface proportion of each secondary structure. The surface exposure degree was defined by the SAS divided by the volume of protein.

The B-factor, which is also called the Debye–Waller factor, indicates the thermal motion-induced attenuation of X-ray scattering or coherent neutron scattering [28,29]. Equation (1) defines the B-factor: (1)$$B=8\pi^{2}<u^{2}>$$
where u (Å) denotes the mean displacement of a scattering center. The B-factor is used to interpret properties such as the thermostability, flexibility, internal motion, and binding of proteins [30,31,32,33,34]. In Alphafold2 models, the B-factor columns are replaced by pLDDT values, which can provide insights into structural flexibility [23]. We converted pLDDT values into pseudo-B-factors since pLDDT values and original B-factors show a reverse relationship. The pLDDT values were first converted into root mean square deviation (RMSD) using the following empirical formula (Equation (2)): (2)$$∆=1.5exp\left[4\left(0.5-pLDDT\right)\right],$$
where ∆ denotes error estimates. pLDDT values were transformed into the scale of 0–1 from the scale of 0–100 [35,36,37]. Then, the converted pseudo-B-factor is expressed as Equation (3) after substituting the converted error estimates into Equation (1), considering the root mean square positional variation in three dimensions.
(3)$$\;B=\frac{8\pi^{2}{∆}^{2}}{3}$$

The converted pseudo-B-factors were calculated for each residue in the proteins. However, in the case of X-ray analysis, low resolution leads to high B-factors around 100–200, and such high values of B-factors are not recommended for making specific conclusions [38]. Therefore, only surface residues with an RMSD smaller than or equal to 1.5 (almost equivalent to *B* ≤ 60) were included in the analysis of surface B-factors. Finally, B-factors were normalized using Equation (4) since a non-normalized B-factor does not represent an absolute quantity and thus cannot be used to compare different protein structures [39]:(4)$$B_{norm}=\frac{B-<B>}{\sigma }$$
where *<B>* denotes the average B-factor in the whole protein structure and *σ* indicates the standard deviation. Then, the mean value of the normalized surface B-factors in a protein was used to characterize the protein surface.

Surface roughness, which can be quantitatively characterized by the fractal dimension (*FD*), was calculated to identify the surface structural irregularity (Equation (5)) [26]: (5)$$FD=2-\frac{dlog\left(A_{s}\right)}{dlog(R)}$$
where $A_{s}$ and R represent the molecular surface area and rolling probe radius, respectively. *FD* falls within the range of 2 to 3, having the smoothest surface at 2 and having the roughest surface at 3. For the calculation of $A_{s}$, we calculated the SES using the 3V calculator (http://3vee.molmovdb.org, accessed on 13 December 2023) [40]. Then, Equation (5) was transformed into Equation (6) for the convenience of calculation.
(6)$$D_{i}=2-\frac{({\log A_{ses})}_{i}-{(\log A_{ses})}_{i-1}}{{(\log\;R)}_{i}-{(\log\;R)}_{i-1}},\;FD=\frac{1}{N}\sum_{n=1}^{N}D_{n}$$
where *i* refers to a probe radius starting from 1.2, in the range of 1.0 to 3.6, with the interval of 0.2 (1.0, 1.2, 1.4, 1.6, …, 3.6, *N* (number of sets) = 13). *i*−1 refers to the previous step of *i* (*i*−1 starts from 1.0). $({\log A_{ses})}_{i}$ indicates the log value of the solvent-excluded surface area under the probe radius *i*. The range of the probe radius is suitable for the analysis since the probe sizes are sensitive to specific interactions between residues, reflecting the size of water molecules and side chains [26]. Finally, the mean value of all the calculated *D_i_* represents the *FD*.

### 2.3. Application of Machine Learning

The logistic regression (LR) model, a regression model for binary classification problems, shows its chief advantage by providing high model interpretability. An odds ratio of each independent variable enables a quantitative evaluation its contribution to dependent variables. Surface descriptors were given as independent continuous variables, and <HAC:1, Extracellular:0 > tags were provided as dependent dichotomous variables in the models. Then, Equation (7) was used to represent the probability of being an HAC protein under the given independent variables [41]:(7)$$P(y=1|x_{1},\;x2,\;\ldots ,\;xi)=\frac{\exp\left[f\left(X_{i},\beta_{i}\right)\right]}{1+\exp\left[f\left(X_{i},\beta_{i}\right)\right]}=\frac{e^{\beta_{0}+{\beta_{1}X}_{1}+{\beta_{2}X}_{2}+\ldots +{\beta_{i}X}_{i}}}{1+e^{\beta_{0}+{\beta_{1}X}_{1}+{\beta_{2}X}_{2}+\ldots +{\beta_{i}X}_{i}}}$$
where *P, x_i_,* and *β_i_* denote the probability of being an HAC protein, a surface descriptor, and an accompanying beta coefficient. LR uses the maximum likelihood method to estimate $\beta_{i}$, and the odds ratio corresponds to *exp*[*β_i_*]. Then, a logistic transformation, which converts the non-linear relationship into the original linear regression equation, is applied as Equation (8).
(8)$$ln\;P=ln\frac{\exp\left[f\left(X_{i},\beta_{i}\right)\right]}{1+\exp\left[f\left(X_{i},\beta_{i}\right)\right]}=ln\frac{P}{1-P}=\beta_{0}+\sum \beta_{i}X_{i}=\beta_{0}+\beta_{1}X_{1}+\beta_{2}X_{2}+\ldots +\beta_{i}X_{i}$$

A positive $\beta_{i}$ indicates that an increase in *x_i_* leads to a stochastic increase in the probability of being an HAC protein. Conversely, a negative $\beta_{i}$ means that an increase in *x*_i_ results in a stochastic decrease in the probability of being an HAC protein. 

As a parametric model, LR requires several statistical assumptions to perform well [41]. Thus, several data preprocessing steps were conducted, including checking the multicollinearity of surface descriptors, deleting strongly influential outliers, and data scaling to meet the assumptions and enhance the model performance. Pearson correlation (PC) analysis, a statistical test that measures the linear association between two variables, was conducted to limit the multicollinearity problem. Also, Cook’s distance from the statsmodels module in Python was calculated for leverage and residual values analysis. Conclusively, 1.03% of the proteins turned out to be highly influential and outliers simultaneously and were thus eliminated from the dataset. Finally, the surface descriptors were standardized with the StandardScaler function in the Python sci-kit learn library for data scaling. 

Upon constructing the LR model, several popular supervised learning algorithms for classifications, including K-Nearest Neighbor (KNN), Random Forest (RF), and Support Vector Machine (SVM), were used to compare the performance of different models. All the algorithms were performed using the Scikit-learn Package in Python 3.9.12. The hyperparameters for each algorithm were optimized using GridSearch cross-validation (CV), where every parameter combination was tested to evaluate the ML models. Five-fold cross-validation was used to avoid overfitting to the test set. Before constructing the machine learning models, the datasets were randomly divided into a training set (80%) and a test set (20%), maintaining the original ratio of the target class. Then, the performance of different models was assessed by predictive indicators including the classification accuracy and the area under the curve of receiver operating characteristic (AUC-ROC) curve. We randomly split the training and test sets five times to avoid sampling bias and overfitting and then reported the mean accuracy of each model. We selected the final ML model, LR, for the feature importance analysis considering its high accuracy and model interpretability. Finally, each descriptor’s significance and importance were explained with statistical analysis.

## 3. Results and Discussion

### 3.1. Pearson Correlation (PC) Analysis

First, PC analysis for all the descriptors in the training set was conducted before applying machine learning. Table 2 shows the PC coefficients among the independent variables, i.e., surface descriptors and dependent variables (where HAC is tagged as 1 and extracellular as 0). A PC coefficient ranges from −1 to 1, showing a perfectly negative correlation at −1 and a perfectly positive correlation at 1. A PC coefficient of 0 represents the absence of a linear correlation. As a result, all the relationships between each surface descriptor and dependent variable were significant at 0.05 (*p* < 0.05) except for the structure surface exposure degree (*s_sf*) (Table 2).

As shown in Figure 3, two descriptors, including the proportion of surface alpha-helices (*s_ah*) and the proportion of total charged surface area (*s_charge_avg*), were highly linearly correlated with the descriptors in their categories including protein structures and charge, respectively. Therefore, the descriptors were eliminated from the descriptor pool, considering that they showed the highest linear correlation with other descriptors in their category. According to the above results, we excluded three descriptors using PC analysis including *s_sf*, *s_ah*, and *s_charge_avg* from the initial pool of ten surface descriptors, thus only applying seven descriptors (s_phobic_avg, s_pos_area_avg, s_neg_area_avg, norm_s_b, s_bs, s_do, and FD) for machine learning.

### 3.2. Comparison of Supervised Machine Learning Algorithms for Binary Classification Problem

The performance of different machine learning algorithms for the binary classification problem (KNN, LR, RF, and SVM) was compared using identical training and test data sets. The performance of each model was evaluated using accuracy and AUC-ROC graphs. The models were compared by randomly splitting the training and test sets five times to avoid the effect of fluctuation in the results (Figure 4a). As a result, all the algorithms showed excellent and similar performance, exhibiting 79.7%, 80.2%, 79.3%, and 80.2% accuracy for KNN, LR, RF, and SVM, respectively. The ROC curves for the algorithms were also in nearly identical and impartial shapes (Figure 4b). The algorithms also demonstrated comparable AUC scores, with the LR exhibiting the highest AUC score (87.6%), albeit not significantly outperforming the other algorithms (87.5%, 87.3%, and 87.1% for KNN, RF, and SVM, respectively). After comprehensively considering prediction performance and interpretability, we chose LR for the feature importance analysis of the surface descriptors.

### 3.3. Results of the Logistic Regression Analysis

Table 3 and Figure 5 show the influence of each surface descriptor on the logistic regression analysis. The coefficients and standard errors of the descriptors were calculated based on the mean values from five randomly split training sets. Table 3 shows that all the surface descriptors are statistically significant at 0.05 (*p* < 0.05). The sign of the coefficient for each descriptor determines its influence on the probability of the protein being classified as an HAC protein: a positive coefficient suggests that an increase in the descriptor value increases the likelihood of the protein being classified as an HAC protein. In contrast, a negative coefficient indicates that an increase in the descriptor value decreases the probability of the protein being classified as an HAC protein. Two descriptors related to surface charge had positive coefficients in the model including the negatively charged surface area (*s_neg_area*) and the positively charged surface area (*s_pos_area*).

On the other hand, the other descriptors including surface hydrophobicity (*s_phobic_avg*), the normalized surface B-factor (*norm_s_b*), the proportion of surface beta structures (*s_bs*), surface roughness (*FD*), and the proportion of surface disordered regions (*s_do*) exhibited negative coefficients. Moreover, the odds ratio, which is the exponentiated coefficient of a descriptor, along with its 95% confidence interval (C.I.), can aid in interpreting each coefficient by providing information on the probability of being an HAC protein [41]. All the statistical summaries of each descriptor are provided in Appendix A. The following sections will provide further statistical details for each descriptor, including their relationships with several issues related to crowded cellular environments and nonspecific interactions.

### 3.4. Proper Folding of HAC Proteins Can Be Achieved with Low Surface Hydrophobicity and Secondary Structure Compositions

Our findings corroborate that HAC proteins adopt a protein folding strategy, limiting nonspecific interactions in crowded environments. A protein entropically prefers a compactly folded state over an unfolded or expanded state in macromolecular crowded environments [42,43,44]. In particular, hydrophobic interactions play a central role in protein folding, clustering non-polar residues in the protein core to form globular structures [45]. On the other hand, polar residues are often exposed to the protein surface, restricting hydrophobic interactions involved in molecular recognition. We observed that the surfaces of HAC proteins exhibited lower hydrophobicity and well-folded states with a lower proportion of disordered regions (Figure 6a,b).

Surface hydrophobicity, as quantitatively measured using the normalized consensus hydrophobicity scale proposed by Eisenberg et al. (Figure 6c) [27], had the highest influence (*s_phobic_avg* = −0.807) among all the surface descriptors (Figure 5). With the considerably high population of highly hydrophilic aspartic acid (D) and arginine (R), we assume that the significantly high surface hydrophilicity on HAC proteins mainly derives from the remarkable scarcity of leucine (L) and notably abundant lysine (K) and glutamic acid (E) (Figure 6d). Our observations of the high population of K and E on the HAC protein surfaces are consistent with the findings of White et al. [9]. Their study demonstrated that molecular chaperones, which require non-adhesive surfaces for reversible interactions with multiple proteins, have a higher abundance of E and K, which possess strong water-binding properties and weak associations with surrounding amino acids. Here, we suggest that highly hydrophobic L also plays a vital role in forming hydrophilic surfaces. While the proportion of L is similar in buried regions of both protein types, there is a significant contrast on the surface region, where HAC proteins are strikingly lacking L compared with extracellular proteins (Figure 6d). Hence, HAC proteins can have a stable hydrophobic core and exhibit higher surface hydrophilicity.

The negative coefficients (*s_bs* = −0.286 and *s_do* = −0.138) shown in Figure 5 indicate that the HAC proteins generally exhibited higher proportions of alpha-helices and lower proportions of beta structures and disordered regions than those of extracellular proteins in both the surface and buried regions (Figure 6b). This trend in surface secondary structures aligns with the global secondary structures of cytoplasmic proteins proposed by Loos et al., which revealed that cytoplasmic proteins are globally more enriched in alpha-helices and show a lower frequency of beta structures and disordered regions [8]. Furthermore, the surface trend in the two well-folded structures, i.e., the alpha-helices and beta structures, can be supported by the previous study by Bhattacharjee and Biswas, which suggested that beta sheets are highly hydrophobic and buried in the core of proteins. In contrast, long polar residues contribute to the formation of alpha-helices [46]. The lower proportions of the disordered regions of the HAC proteins can be explained by the nonspecific interaction propensity of its innate flexibility. The study by Nishizawa et al. highlighted the engagement of disordered regions in nonspecific interaction, observing the nonspecific ATP–protein interactions in intrinsically disordered proteins and flexible regions [47]. Their study used NMR spectroscopy and molecular dynamics simulations to capture concentration-dependent noncovalent interactions between ATP and disparate proteins. As a result, the interaction was notably distinct in the intrinsically disordered proteins (α-synuclein) and flexible regions (loops or termini). Our findings regarding the hydrophobicity and secondary structures on the surfaces of HAC proteins support the protein folding strategy for environmental adaptation in crowded environments. 

### 3.5. HAC Proteins Are Emphasized with Surface Rigidity and an Extreme Range of Net Surface Charge

HAC proteins should have different structural surface characteristics to function correctly in a crowded environment. For instance, proteins in cellular environments are expected to have better thermostability with higher melting temperatures due to the crowding effect [3]. Previous studies have shown that increased thermostability is often accompanied by a decreased overall flexibility of proteins [30,31]. Also, protein solubility, which indicates the characteristic of a protein to maintain its intact state, is an essential issue for protein stability to avoid aggregation, which refers to protein binding accompanying irreversible conformation change [48]. Here, we plotted the distributions of surface pseudo-B-factors and the distributions of surface charges to understand the surface flexibility and stability of HAC proteins (Figure 7).

A pseudo-B-factor increases as protein structures show more considerable flexibility [23]. We obtained two insights from Figure 7a: (1) the surfaces of HAC proteins tend to have lower flexibility than extracellular proteins and (2) the lower flexibility on the surfaces of HAC proteins is emphasized as the analyzed domain is shifted from buried regions to surface regions. The lower flexibility on the surfaces of an HAC may be supported by recent findings on the direct relationship between protein intracellular abundance and thermal stability, which is often observed with reduced flexibility [49,50]. The findings showed that the protein interface stability was positively correlated with the protein abundance, enabling the prevention of misinteractions. At the same time, abundant intracellular proteins with high thermostability were less prone to aggregation or local unfolding. Thus, we suggest that the surfaces of HAC proteins reflect reduced flexibility to be adaptive in crowded environments. 

Two charge-related descriptors with positive coefficients contributed to the model with nearly equivalent scales (*s_neg_area* = 0.622, *s_pos_area* = 0.617) (Figure 5). Our findings show that the richness of both negatively charged and positively charged areas is significant on the surfaces of HAC proteins compared with extracellular proteins (Figure 7b). To further understand the charge distribution on protein surfaces, we plotted the net surface charge distribution of extracellular and HAC proteins using the rearranged Henderson–Hasselbalch equation (for more details, see Appendix A) (Figure 7c) [51,52]. In nature, it is known that zwitterionic surfaces with evenly distributed positively and negatively charged residues help resist nonspecific interactions with stronger hydrostatic repulsion fields [4]. Our data showed the more extreme range of net surface charge in HAC proteins. We assume that the results come from the complex considerations of aggregation and solubility. For instance, Ryan et al. elucidated that increased protein solubility is strongly correlated with negative surface charge, explained by the water-binding properties of E and D [53]. Also, positively charged amino acids like K and R have effectively inhibited aggregation by weakening protein–protein interactions [54]. To sum up, our results showed a higher charged surface and extreme net charge range on the surfaces of HAC proteins, and we assume that this was the result of complex behaviors of HAC proteins for adaptation in a crowded environment.

### 3.6. The Smoother Surface of HAC Proteins May Modulate Molecular Adsorption

As mentioned, molecular crowding and protein abundance are crucial for studying nonspecific interactions. We hypothesized that the surface geometry of HAC proteins should have strategies for minimizing molecular adsorption and nonspecific interactions. Surface roughness is a critical parameter used to describe surface geometry. Indeed, nano-scale surface roughness was found to have a significant influence on protein–protein interactions [55,56]. Also, surface homogeneity and low surface roughness were found on the surface of streptavidin, which is known to have exceptionally strong specific binding with biotin and exhibits low nonspecific binding [57]. Here, we calculated the surface roughness of proteins using *FD*, which can represent the degree of surface irregularity [26]. *FD* shows the lowest value for a completely smooth surface (*FD* = 2). In contrast, it has the highest value for the roughest protein surface (*FD* = 3). With *FD* of all proteins ranging from 2.044 to 2.372, we observed subtle but discernable distinctions between the extracellular and HAC proteins (Figure 8). 

The HAC proteins exhibited smoother surfaces in general, which can be inferred by the large population of Alanine, which has the shortest residue chain length among 20 amino acids (Figure 6d). In addition, among four types of aromatic amino acids (Tryptophan, Phenylalanine, Tyrosine, and Histidine) that can have higher van der Waals volumes, three of them (Tryptophan, Phenylalanine, and Histidine) were more abundant on the surfaces of extracellular proteins. Considering that protein surface roughness is necessary upon binding with small molecules [58], we suggest that the smoother surface of an HAC protein can be a strategy for minimizing small molecules-induced nonspecific interactions. However, further investigation will be necessary to substantiate our assumptions.

## 4. Summary and Conclusions

In this study, we utilized surface physicochemical, structural, and geometrical descriptors to identify HAC proteins with ML and quantitatively analyzed the surface characteristics. We first solved binary classification for HAC and extracellular proteins using several supervised ML algorithms (KNN, LR, RF, and SVM). Then, LR was chosen for the descriptors’ final feature importance analysis, considering both excellent model performance (80.2% accuracy, 87.6% AUC) and high model interpretability. The charge-related descriptors showed positive correlations, while hydrophobicity, the B-factor, the proportion of beta structures, roughness, and the proportion of disordered regions exhibited negative correlations with the HAC proteins in the importance analysis of descriptors.

We also found that the E, K, and L populations and well-folded secondary structures on the HAC protein surfaces played vital roles in their hydrophilicity and compactly folded structures. Also, we observed limited protein flexibility and extreme net charge from the surfaces of HAC proteins, which previous studies on the adaptation of cytoplasmic proteins in crowded environments can explain. Finally, we suggested that smoother surfaces of proteins can be critical in minimizing the nonspecific adsorption of small molecules. Our results indicate that several surface descriptors can be employed to identify, quantify, and explain protein surface characteristics in a crowded cellular environment.

To summarize, our study primarily shows the combinatorial impact of surface descriptors with disparate properties in characterizing HAC proteins and distinguishing them from extracellular proteins with ML-based approaches. At the same time, it is important to note that our findings are subject to certain limitations, such as determining an optimal threshold for pLDDT values and incorporating multimeric protein structures.

Our findings on the quantitative analysis of the descriptors could facilitate the design of surfaces that are well-adapted to crowded environments, such as nonspecific interaction-resistant surfaces with selectivity to target materials [59,60,61,62,63,64]. One example of the application is the design of immunosensors, where the nonspecific adsorption of various biomolecules causes background noise and critically impairs sensitivity [65]. Another field highlighting the importance of nonspecific interaction-resistant surfaces is reducing protein corona on nanoparticles [66]. When nanoparticles first come into contact with biological fluid, proteins attach to their surfaces and form a protein layer, i.e., protein corona. Since protein corona causes direct impacts on the performance of nanoparticles, the new strategy—applying a nonspecific interaction-resistant surface—for nanoparticles should aim to reduce or slow protein corona formation. 

## Figures and Tables

**Figure 1 biomimetics-09-00162-f001:**
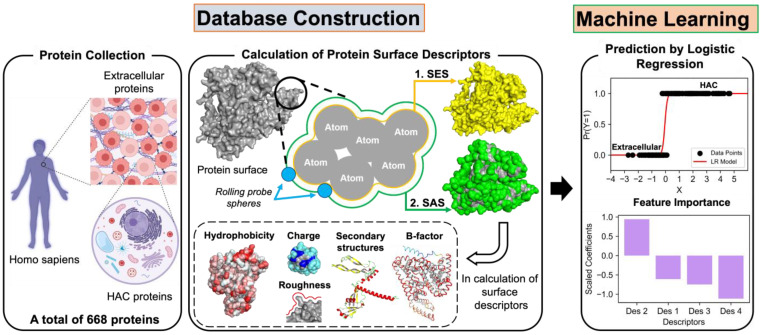
Schematic representation of key processes in the functional prediction and quantitative analysis of surface physicochemical, structural, and geometrical descriptors on protein surfaces. HAC: highly abundant cytoplasmic; SES: solvent-excluded surface; SAS: solvent accessible surface.

**Figure 2 biomimetics-09-00162-f002:**
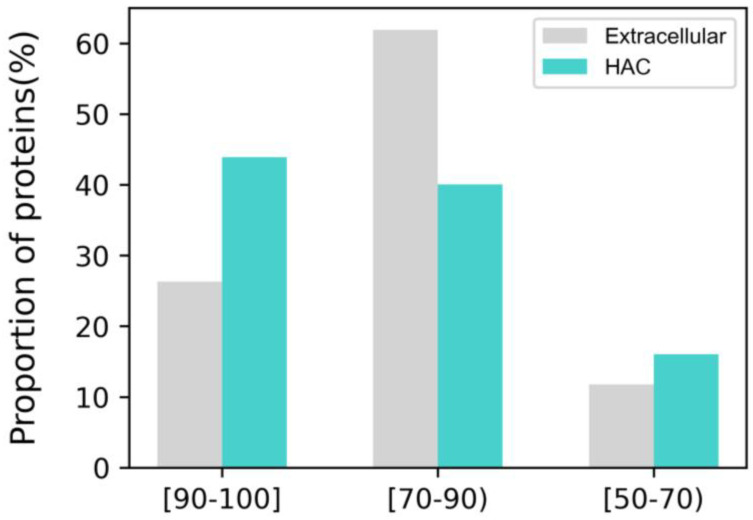
Average predicted local distance difference test value distribution of the 668 collected proteins.

**Figure 3 biomimetics-09-00162-f003:**
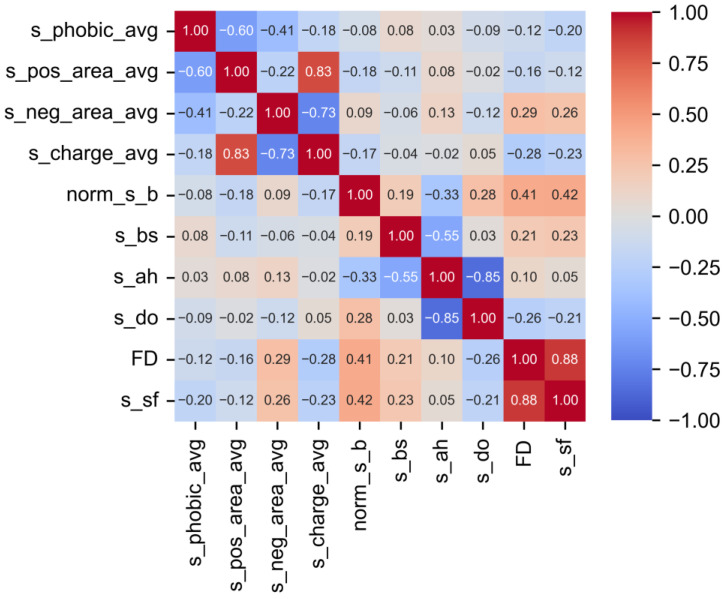
PC coefficients among the descriptors from the training set.

**Figure 4 biomimetics-09-00162-f004:**
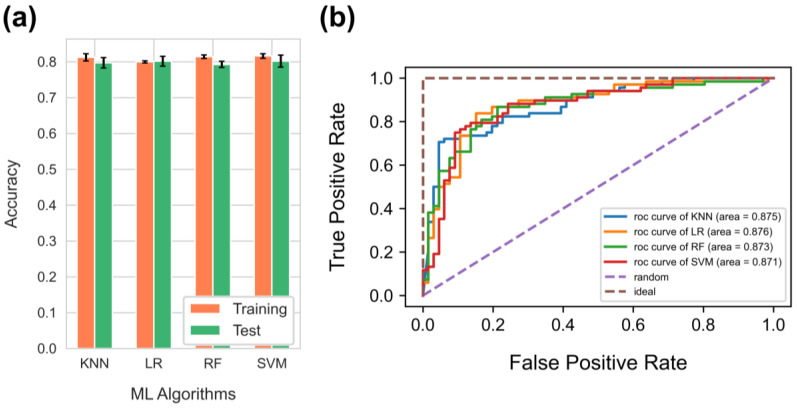
(**a**) Comparison of the performance of different ML algorithms. KNN: K-Nearest Neighbor; RF: Random Forest; LR: logistic regression; and SVM: Support Vector Machine. (**b**) ROC curves for the four machine learning algorithms from a single-shot trial. The hyperparameters used to tune each model in a single-shot trial are described in Appendix A.

**Figure 5 biomimetics-09-00162-f005:**
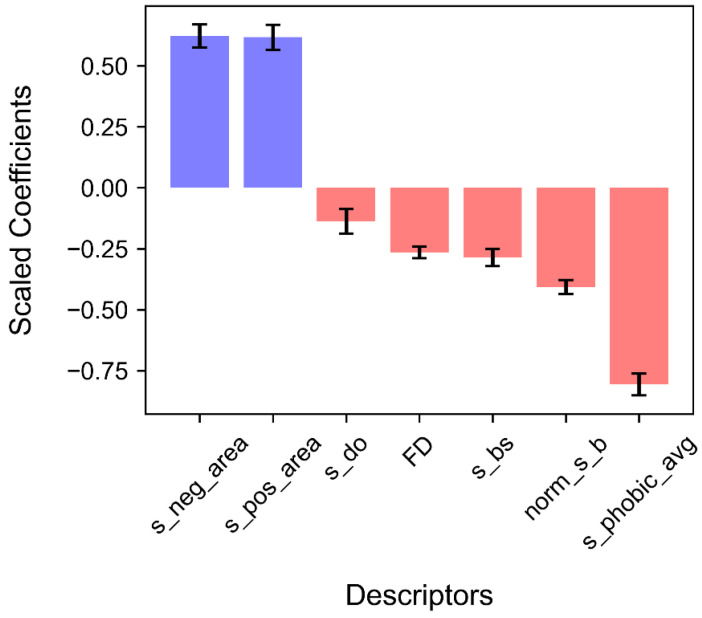
Importance of surface descriptors in classifying proteins into HAC and extracellular proteins. Error bars denote the standard deviation of five randomly split training and test sets, performed to prevent sampling bias and overfitting.

**Figure 6 biomimetics-09-00162-f006:**
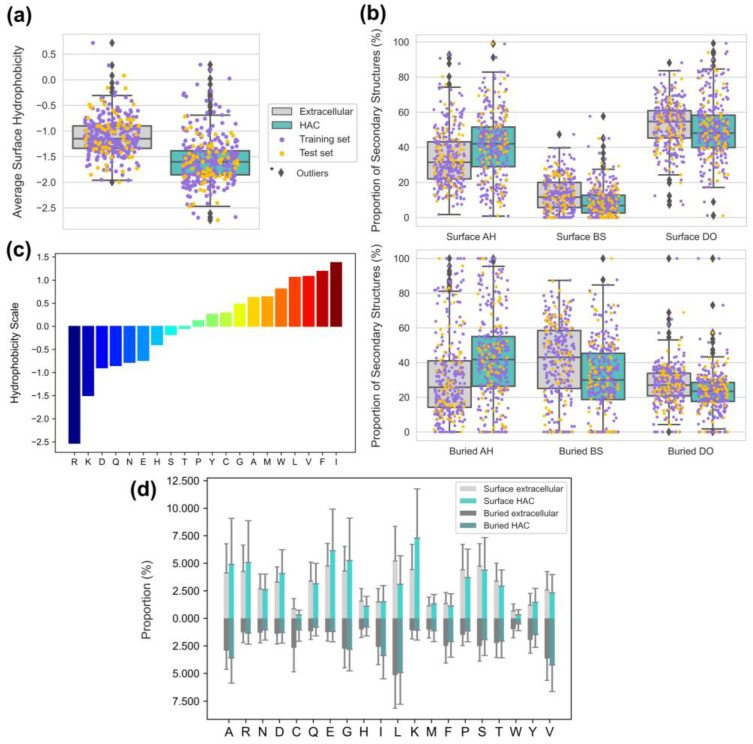
(**a**) Boxplots of the average surface hydrophobicity of extracellular and HAC proteins. (**b**) Boxplots of the proportion of secondary structures of the extracellular and HAC proteins (AH: alpha-helix; BS: beta structure; and DO: disordered region) in the surface and buried regions. (**c**) Hydrophobicity scale of 20 kinds of amino acids. (**d**) Proportion of amino acids in the surface and buried regions of proteins.

**Figure 7 biomimetics-09-00162-f007:**
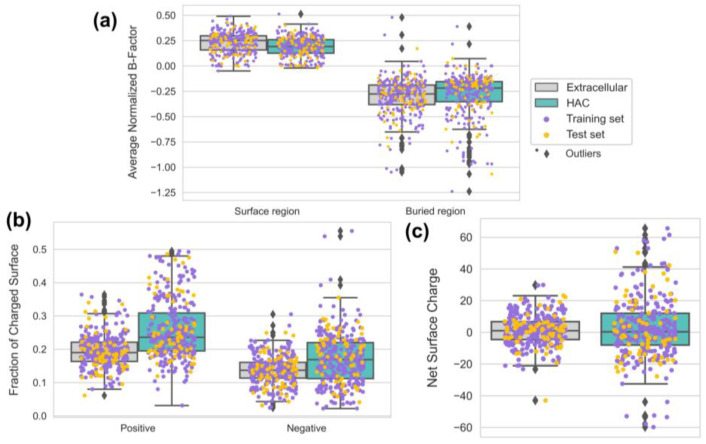
Boxplots of (**a**) the average surface B-factors on the surface and buried regions, (**b**) the fraction of positively charged and negatively charged surface area on extracellular and HAC proteins, and (**c**) the net surface charge of extracellular and HAC Proteins.

**Figure 8 biomimetics-09-00162-f008:**
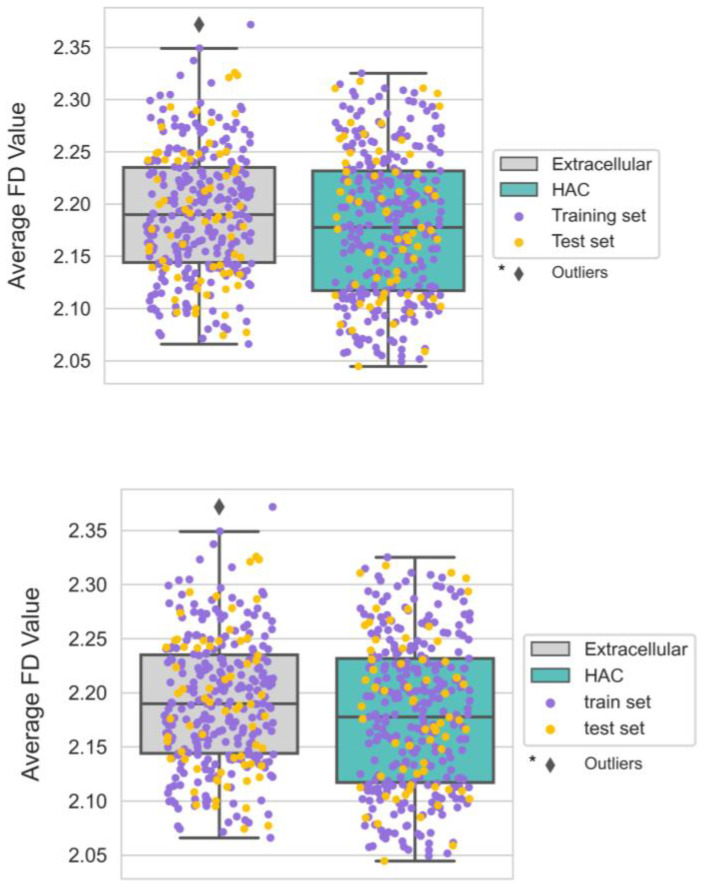
Boxplots of the surface roughness of extracellular and HAC proteins.

**Table 1 biomimetics-09-00162-t001:** Descriptors used in this work to explain protein surface characteristics.

Category	Variables (Descriptors)	Definition	Analyzed Surface
Hydrophobicity	*s_phobic_avg*	Average surface hydrophobicity	Solvent-accessible surface
Charge	*s_pos_area*	Fraction of positively charged surface area	
	*s_neg_area*	Fraction of negatively charged surface area	
	*s_charge_avg*	Fraction of total charged surface area	
Protein structure	*s_ah*	Proportion of surface alpha-helices	
	*s_bs*	Proportion of surface beta structures	
	*s_do*	Proportion of surface-disordered regions	
	*s_sf*	Structure surface exposure degree	
Flexibility	*norm_s_b*	Average normalized surface B-factors	
Geometry	*FD*	Average protein surface roughness	Solvent-excluded surface

**Table 2 biomimetics-09-00162-t002:** Pearson Correlation (PC) Coefficients between independent and dependent variables from a train set.

Surface Descriptors		PC Coefficient
Hydrophobicity	*s_phobic_avg*	−0.472 **
Charge	*s_pos_area*	0.401 **
	*s_neg_area*	0.239 **
	*s_charge_avg*	0.142 **
Protein structures	*s_ah*	0.206 **
	*s_bs*	−0.228 **
	*s_do*	−0.102 *
	*s_sf*	−0.023
Flexbility	*norm_s_b*	−0.225 **
Geometry	*FD*	−0.106 *

** *p*-value < 0.01, * *p*-value < 0.05.

**Table 3 biomimetics-09-00162-t003:** Results of the logistic regression analysis for each surface descriptor.

Logistic Regression Analysis							
Descriptor	β	S.E.	*z*-Value	Significance Level	Odds Ratio	Exp(β) 95% C.I.	
						Min	Min
*s_phobic_avg*	−0.807	0.045	−17.913	<0.001	0.446	0.408	0.487
*s_pos_area_avg*	0.617	0.051	12.016	<0.001	1.853	1.675	2.049
*s_neg_area_avg*	0.622	0.047	13.112	<0.001	1.862	1.697	2.043
*norm_s_b*	−0.408	0.029	−13.972	<0.001	0.665	0.628	0.704
*s_bs*	−0.286	0.036	−7.992	<0.001	0.751	0.700	0.806
*s_do*	−0.138	0.050	−2.738	<0.05	0.872	0.790	0.962
*FD*	−0.265	0.024	−11.211	<0.001	0.767	0.733	0.804

## Data Availability

Data are contained within the article and Appendix A.

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
