# Peer review of "Application of Machine Learning in the Quantitative Analysis of the Surface Characteristics of Highly Abundant Cytoplasmic Proteins: Toward AI-Based Biomimetics"

_biomimetics, 2024, doi:10.3390/biomimetics9030162_

Round 1

Reviewer 1 Report

Comments and Suggestions for Authors

I carefully read the manuscript submitted by Joo A Moon et al. The authors utilized surface physicochemical, structural, and geometrical descriptors to identify highly abundant cytoplasmic (HAC) proteins with machine learning (ML) and quantitatively analyzed the surface characteristics.

The authors tackle the essential problem of nonspecific protein-protein interactions (PPIs) and differentiate highly abundant cytoplasmic proteins from extracellular proteins. To predict specific PPIs, homology modeling as well as ab initio docking methods are long-standing and have been extensively studied (https://doi.org/10.1002/prot.25889 & https://doi.org/10.1016/j.tibs.2023.03.003 ). However, distinguishing specific and non-specific PPIs without any prior knowledge remains a challenge for the scientific community. Thus, the study is important in view of identifying HAC proteins from those of extracellular proteins using binary classification algorithms based on different surface descriptors.

As stated in the above fields the manuscript, although not extremely novel, is clear, concise, well-written, and appears to be interesting to those working in the field of protein structures and their interactions. Methods presented in this study use 3D structures collected through the Alpha-fold protein structure database. The two different datasets, human cytoplasmic proteins with high abundance and extracellular proteins, were considered and surface descriptors were calculated for each data set. The logistic regression model which is straightforward but useful for binary classification was applied by dividing into training and test data. The results achieved via ML are well-discussed in the main body of the reported manuscript. There are a few points that should be addressed before considering this work for publication. Here are some specific remarks: 

In Figure 4 it’s not clear what surface descriptor was used in classification. In sections 4.4 and 4.5, the distribution of surface descriptors for two different data sets overlaps although the average values are different. In my opinion, it is a standard method with no drastic improvement over other classification methods based on surface descriptors. So, what warrants it being published here? Should be revised in the conclusion. Furthermore, It would be useful if the data set were separated based on AlphaFold model accuracy (pLDDT score) and see if the accuracy improves for high-accuracy models.  

Reviewer 2 Report

Comments and Suggestions for Authors

The goal of this study is to find out what surface properties cytoplasmic proteins have that enable them to exist in a crowded environment. The authors compared a set of highly abundant cytoplasmic and a set of extracellular proteins with regard to a number of surface descriptors. The methodology is sound and the conclusions are supported by the results. I have the following comments and suggestions:

1. The list of proteins (preferably Uniprot names e.g. G3P_HUMAN) should be published in the supplementary material.

2. It would be useful to see the length distribution of proteins in both groups, as it might differ between the groups and this may relevant for the findings.

3. Fig. 1 says "Total number around 670". Please write the exact number (668)

4. Line 47, please clarify what "global regions" means.

5. Section 2 is followed by section 4 in the manuscript; there is no section 3. Please fix.

6. Equation 6 is not compatible with equation 5, as eq. 5 contains a subtraction from 2 while eq. 6 does not. Please fix.

7. Based on Fig. 7a, the difference in B-factors between the two groups is negligible (definitely not statistically significant). The text seems to exaggerate the difference.

8. Please comment on the limitation that the Alphafold models were only built for single chains while many proteins may be oligomers of multiple chains (homo- or heterooligomers).

9. It is not entirely clear why only the high-abundance cytoplasmic proteins were used for this study. The cytoplasmic proteins with lower abundance are in an equally crowded environment. Does the abundance have a role here? Please comment.

Comments on the Quality of English Language

English is good.

Round 2

Reviewer 1 Report

Comments and Suggestions for Authors

In the revised manuscript, the authors responded to all comments and revised the text accordingly.